# Engagement, Advance Care Planning, and Hospice Use in a Telephonic Nurse-Led Palliative Care Program for Persons Living with Advanced Cancer

**DOI:** 10.3390/cancers15082310

**Published:** 2023-04-15

**Authors:** Rebecca Liddicoat Yamarik, Laraine Ann Chiu, Mara Flannery, Kaitlyn Van Allen, Oluwaseun Adeyemi, Allison M. Cuthel, Abraham A. Brody, Keith S. Goldfeld, Deborah Schrag, Corita R. Grudzen

**Affiliations:** 1Long Beach Veterans Affairs, Long Beach, CA 90822, USA; 2Ronald O. Perelman Department of Emergency Medicine, New York University Grossman School of Medicine, New York, NY 10016, USA; 3Rory Meyers College of Nursing, New York University, New York, NY 10010, USA; 4Division of Geriatric Medicine and Palliative Care, New York University Grossman School of Medicine, New York, NY 10016, USA; 5Department of Population Health, New York University Grossman School of Medicine, New York, NY 10016, USA; 6Memorial Sloan Kettering Cancer Center, New York, NY 10065, USA

**Keywords:** nurse-led, telephonic, palliative care

## Abstract

**Simple Summary:**

Persons living with advanced cancer have significant symptoms and psychosocial needs that often result in visits to the Emergency Department. We enrolled persons living with advanced cancer in a 6-month program to receive telephone calls from nurses to help manage their symptoms, coordinate care, and explore their values and goals for future care (advance care planning). About half of the subjects completed the 6-month program, a quarter died or enrolled in hospice, 19% were lost to follow-up, and 9% withdrew from the program. White patients and those with fewer symptoms were more likely to withdraw. Eighty-three percent of all patients completed some advance care planning, and 80% of patients received hospice care prior to death.

**Abstract:**

Persons living with advanced cancer have intensive symptoms and psychosocial needs that often result in visits to the Emergency Department (ED). We report on program engagement, advance care planning (ACP), and hospice use for a 6-month longitudinal nurse-led, telephonic palliative care intervention for patients with advanced cancer as part of a larger randomized trial. Patients 50 years and older with metastatic solid tumors were recruited from 18 EDs and randomized to receive nursing calls focused on ACP, symptom management, and care coordination or specialty outpatient palliative care (ClinicialTrials.gov: NCT03325985). One hundred and five (50%) graduated from the 6-month program, 54 (26%) died or enrolled in hospice, 40 (19%) were lost to follow-up, and 19 (9%) withdrew prior to program completion. In a Cox proportional hazard regression, withdrawn subjects were more likely to be white and have a low symptom burden compared to those who did not withdraw. Two hundred eighteen persons living with advanced cancer were enrolled in the nursing arm, and 182 of those (83%) completed some ACP. Of the subjects who died, 43/54 (80%) enrolled in hospice. Our program demonstrated high rates of engagement, ACP, and hospice enrollment. Enrolling subjects with a high symptom burden may result in even greater program engagement.

## 1. Background

Persons living with advanced cancer have high symptom burden, low quality of life, and high emotional and spiritual needs compared to those with early-stage cancer [1]. The symptom and psychological support needs of this population are often greater than what is available in a traditional oncology clinic setting. Multiple symptoms, including fatigue, pain, dyspnea, anorexia, constipation, depression, and anxiety, make a standard 15-min clinical encounter insufficient to address complex needs [2]. This can result in visits to Emergency Departments (EDs) for symptoms that are manifestations of cancer and/or its treatment, and these patients often benefit from more frequent follow-up and care coordination. Guidelines from the American Society of Clinical Oncology recommend that persons living with advanced cancer receive specialist palliative care from diagnosis [3]. However, workforce shortages of palliative care providers, particularly physicians and advanced practice providers, limit the more widespread provision of specialty palliative care alongside traditional cancer care [4,5].

Nurses can augment traditional specialist palliative care using one of several existing palliative care delivery models. For instance, support lines staffed by oncology nurses who respond to calls for concerning symptoms, changes in clinical status, and medication management have demonstrated improvement in symptom severity, but have not shown reductions in health care use [6,7]. Pre-scheduled, nurse-initiated calls for patients receiving chemotherapy have also demonstrated improvements in symptom management [8]. Some interventions use lay navigators to deliver primary palliative care with a focus on caregivers, while others involve nurses during in-person oncology infusion visits [9,10,11,12]. Several current interventions under exploration include nurses responding to patients’ portal-reported messages and nurse-led telephonic interventions alone or in combination with in-person multidisciplinary sessions [13,14,15]. The interventions are often psychoeducational, where nurses teach problem-solving and address the eight palliative care domains [16]. While engagement in a variety of models has been reported elsewhere, predictors of engagement have been less well studied. Few models have examined impacts on hospice use. This study aims to assess (1) the characteristics of patients that withdrew from the program, (2) the proportions of advance care planning (ACP) processes completed, and (3) the sociodemographic and clinical factors associated with death in hospice among the study population.

## 2. Methods

### 2.1. Study Design and Population

This is a secondary analysis of data from a single intervention arm of persons with advanced cancer enrolled in Emergency Medicine Palliative Care Access (EMPallA), a comparative effectiveness trial testing two forms of palliative care delivery for persons living with serious illness following ED discharge: nurse-led telephonic care versus specialty outpatient palliative care [17]. The analysis only includes patients with advanced cancer that were randomized to the nurse-led telephonic arm of the study between April 2018 and June 2022. Data comparing the two interventions on primary (quality of life) and secondary outcomes (health care use, loneliness, and survival) are forthcoming. For the nurse-led telephonic care arm, registered nurses (RNs) certified in hospice and palliative nursing housed at New York University Grossman School of Medicine (NYUGSOM) delivered telephonic palliative care across 18 sites under the supervision of a hospice and palliative medicine physician. The telephonic intervention arm protocol and processes have been reported elsewhere [18]. This study was approved by NYUGSOM’s Institutional Review Board (ClinicialTrials.gov: Identifier NCT03325985).

### 2.2. Inclusion and Exclusion Criteria

Individuals were eligible if they met the following criteria: diagnosis of a metastatic solid tumor prior to an index ED visit at one of the sites during the study period; plan for ED discharge or observation; 50 years of age or older; English or Spanish speaking; and possess health insurance [17]. Exclusions included: two or more outpatient palliative care visits in the previous 6 months; hospice use; history of dementia; residence in skilled nursing or assisted living facility; no working telephone; or residence outside a pre-defined geographic area. EMPallA patients with end-stage organ failure and those randomized to receive specialty outpatient palliative care were outside the scope of this analysis.

Between April 2018 and June 2022, the EMPallA trial randomized 1284 subjects, with 640 in the nursing arm, 229 of whom had advanced cancer (Figure 1). Of those 229 subjects, five withdrew immediately before the RNs received contact information, three did not meet the criteria, and three were moving to a state outside of the nurses’ licensure immediately after enrollment so could not participate. Thus, this analysis comprised 218 persons living with cancer.

### 2.3. Instruments/Measures

The predictor variables were sociodemographic characteristics (age, sex, race, ethnicity, income, and educational level), quality of life, symptom burden, and loneliness. Quality of life was measured by the Functional Assessment of Cancer Therapy-General (FACT-G, Version 4; Cronbach alpha = 0.88). The FACT-G is a 27-item five-point Likert-type instrument scored from 0 to 108 [19]. Higher scores indicate better quality of life. Symptom burden was measured by the revised Edmonton Symptom Assessment Scale (ESAS-r, Cronbach alpha = 0.87) [20]. The ESAS-r is a 10-item numerical rating scale scored from 0 to 100. Clinical significant symptom burden is defined using a cutoff of 31 or higher. For this study, symptom burden was measured as a binary variable: high (score of 31 or higher) or low [21,22]. Loneliness was measured by the three-item loneliness scale (Cronbach alpha = 0.72) [23]. The three-item loneliness scale is a three-point Likert-type scale scored from three to nine. A score of 7 or higher represents “very lonely” [24]. For this study, loneliness was measured as a binary variable: very lonely or not.

Engagement was measured as time to withdrawal from the 6-month program, with censoring resulting from loss to follow-up, death, or 6 months, whichever came first.

Patients were contacted by nurses weekly to monthly depending on their needs, and ACP process data were collected by nurse interventionists. The elements of ACP included: goals of care conversations; naming, informing, or documenting a healthcare proxy in the electronic health record (EHR); completing an advance directive (AD); sharing the AD with a provider or healthcare proxy; documenting the AD in the EHR; and discussing hospice. We do not know whether the nurse was completing a healthcare proxy or AD form de novo or updating a previously completed document, as this was outside the scope of this analysis. It is also unknown whether patients’ wishes would represent a change from those previously documented.

### 2.4. Analysis

For the nurse-led palliative care intervention group with advanced cancer, we describe baseline sociodemographic and clinical characteristics using means and standard deviations for continuous variables, and frequencies and percentages for categorical variables. To assess predictors associated with time to withdrawal from the telephonic program, we estimated a mixed effects Cox proportional hazards model, adjusting for age, sex, presence of a caregiver, income, race, ethnicity, functional status, religion, education, quality of life and symptom burden at trial enrollment, as well as whether the patient was enrolled before or during the COVID-19 pandemic (Table 1). COVID-19 pandemic status was defined as “present” if the index ED visit was on or after 1 February 2020. The model included a site-specific random effect to account for possible clustering by location or hospital site. All covariates were pre-determined a priori. Additionally, we evaluated the need to include an RN-specific random effect to account for potential clustering at the RN level; we determined that its inclusion did not impact the overall estimates, so this was excluded from the final model. To assess the effect of patient characteristics associated with dying in hospice, we used logistic regression modeling. Observations with missing data at baseline were excluded from the analyses. All analyses were conducted using R, Version 4.1.0 (R Foundation for Statistical Computing), including the survival and lme4 packages [25,26,27].

## 3. Results

The mean age of subjects was 66 (SD: 10) years and 107 (49%) were male (Table 1). The sample was nearly three-quarters white, one-quarter black, and almost exclusively English-speaking; please refer to Table 1 for additional study participant characteristics. The mean FACT-G score at baseline was 68 (IQR: 55 to 83) out of a possible score of 108. Eighty-eight (41%) had low symptom scores and 33 (15%) reported they were very lonely.

There were 211 patients included in the withdrawal analysis because 7 patients had missing covariate information at baseline. Of these 211 subjects, 105 (50%) were engaged for the full 6 months and graduated the program. Fifty-four (26%) died or were enrolled in hospice during the 6-month intervention period. Forty (19%) stopped answering or returning calls and were considered lost to follow-up. Nineteen (9%) actively withdrew from the program, stating they no longer wished to be contacted by the nurse (Figure 1).

Of the 19 subjects who withdrew from the study, six (32%) completed the initial assessment and were engaged for 2 to 4 months, while 13 (68%) withdrew during the initial RN phone call. Of those 19 subjects, 11 were no longer interested in the program, four had too many scheduling conflicts due to work or other doctor appointments, two became incapable of speaking on the phone, and two withdrew after speaking with their oncologist.

Based on the Cox proportional hazards model (Table 2), white subjects were more likely to withdraw than non-whites (HR 3.74, [95% CI: 1.01, 13.8]), and subjects with low baseline symptom burden were more likely to withdraw than those with high symptom burden (HR 1.90 [1.02, 3.53]). Other characteristics, including age, sex, having a caregiver, income, ethnicity, education level, functional status, recruitment pre-COVID-19, and quality of life as measured by the FACT-G did not appear to be related to the decision to withdraw.

Table 3 shows the completion rates for ACP for all 218 subjects. By the end of the 6-month program, 182 (83%) subjects completed at least one aspect of ACP. Naming and/or documenting a healthcare proxy in the EHR was the most common element of ACP that was completed (165, 76%). Twenty-two percent discussed hospice, the least commonly completed element. Nurses were able to engage in two-way communication with oncologists only 29% of the time. Of the patients who died on hospice, 95% (41/43) completed some ACP.

Of the 54 subjects who died, 43 (80%) enrolled in hospice. The median hospice length of stay was 6 days (IQR: 1 to 22). Hospice enrollment was not associated with age, sex, race, ethnicity, education level, or symptom burden (Table 4). Patients who reported lower quality of life at baseline were more likely to enroll in hospice (HR 0.92, [95% CI: 0.80, 0.99]) (Table 4).

## 4. Discussion

In our study of 218 persons living with advanced cancer enrolled from the ED in a telephonic nurse-led palliative care intervention, nearly half remained engaged throughout the 6-month telephonic program, a quarter died, and the rest were lost to follow-up or withdrew. Subjects who were white or had low symptom burden were most likely to actively withdraw. A large majority of subjects who died were enrolled in hospice, although the median length of stay was short at 6 days. ACP completion appeared to be high in subjects who died. Nurses had difficulty engaging with oncologists on behalf of patients.

Our study is one of the few to demonstrate high engagement and high hospice enrollment in an advanced cancer population cared for telephonically. Approximately three-quarters of subjects remained engaged in the program until death, hospice enrollment, or completion of the six-month intervention period—a high level. Most previous nurse-led telephonic interventions for persons living with cancer have focused on symptom management in individuals undergoing chemotherapy, with few focused exclusively on advanced cancer. Several have used interventions that were triggered by patient-reported symptoms via smartphone or web-based apps [8,28]. Looking at other populations, a meta-analysis of nurse-led telehealth for older adults and telerehabilitation reported improved overall quality of life, self-care, and chronic disease indicators [29]. These analyses reference a need for more research to explore which individuals may benefit the most as well as the optimal methods for delivery. Moreover, whether these programs should be time-limited or provided on an ongoing basis to those with serious illness is yet to be determined.

The telephonic arm had a low rate of active withdrawal, similar to other programs for persons living with advanced cancer [30]. Engagement was higher than in the CONNECT trial, where oncology infusion nurses made three in-person or telephonic visits over three months in 56% of subjects and completed at least two visits in 78% [12]. Withdrawal from our program was higher for those with lower symptom burden. It is possible that subjects without burdensome symptoms did not feel the program would be helpful. This aligns with evidence from subjects with low symptom burden enrolled in the ENABLE II trial of telephonic advanced practice nursing care who did not find the program helpful. This led authors to include patients in later cancer stages and with higher symptom burden in subsequent trials [16,31]. Selectively enrolling persons with high symptom burden in telephonic nurse-led programs may improve adherence and engagement and may also more efficiently target scarce nurse resources. Persons who self-identify as white were also more likely to withdraw, which has not been seen elsewhere. The cause is unclear and could be an artifact. One hypothesis is that the program increased access to those who experience systemic racism and bias within the healthcare systems [32,33]. Navigation and access provided by the nurses may be more highly valued by this population. Confirming and investigating this phenomenon would require exploration in future work.

Our study showed high levels of ACP completion and this appeared quite high for subjects who eventually died. Our nurses had training in motivational interviewing, which was shown to increase ACP readiness in a Veterans Affairs study compared to usual care [18,34,35]. Nurses were also trained in Respecting Choices, an established ACP engagement program [36]. It is possible that training nurses in multiple tools to deliver ACP telephonically may improve uptake.

Eighty percent of those who died were enrolled in hospice prior to death, which appears to be high based on other studies of persons living with advanced cancer (68% and 55%, respectively) [31,37]. The median length of stay in hospice was 6 days, lower than the median of 18 days nationally for persons living with cancer [38]. A shorter hospice length of stay is associated with decreased satisfaction among bereaved caregivers [39]. Our short hospice length of stay could be explained by the large percentage of New York residents, a state with one of the shortest lengths of stay and lowest hospice utilization rates in the country [38,39]. More research is needed to evaluate how to improve hospice length of stay using telephonic interventions. Only one telephonic intervention to date has demonstrated reductions in health care use and increased use of hospice [37].

One challenge was that nurses were able to engage in two-way communication with oncologists less than one-third of the time. Studies have shown that oncologist understanding of patient care preferences and prognosis can improve communication around goal-concordant care and increase earlier enrollment in hospice [39,40]. Working within a single health system enables internal communication practices such as electronic health record secure messages and inbox messages that were unavailable to the nurses in this program. The nurses instead attempted to empower patients and their caregivers to initiate conversations with their oncologists; documentation of whether or not this occurred is unknown and outside the scope of this study. Providing multiple methods for communicating prognostic understanding and wishes may improve goal-concordant care. More research is needed to identify optimal methods of communication to improve oncologist engagement with telephonic nurses, especially when they are not housed in the same home institution.

## 5. Strengths and Limitations

A strength of the study was its multi-site design and geographic spread, enhancing the generalizability of our results. While centralizing the nurses created some limitations to communication, it allowed for high fidelity to the intervention, enabled oversight by a single physician, and makes the model more easily scalable as a result. High levels of subject engagement with telephonic care, ACP completion, and hospice use demonstrate some of the potential value of the intervention.

Forthcoming data will compare the two forms of palliative care delivery and provide more robust effectiveness data. In this focus on persons with advanced cancer in the nurse-led telephonic arm, 75% of subjects were still alive at the end of the 6-month program. It is possible that a longer program, enrollment of patients closer to the end of life, or a program that followed subjects through death may demonstrate more benefits. We did see that subjects who died had high ACP completion rates and subjects with higher symptom burden had better engagement. Focusing on those with high symptom burden could better target the population most likely to benefit from future telephonic, nurse-led programs.

Descriptive implementation data such as the number of calls, duration, content of each specific call, and any changes in goals of care or ADs were not collected systematically by all nurses delivering the intervention to all patients. The fidelity of the intervention was monitored qualitatively using program implementation observations. Unlike other studies, our nurses did not deliver specific educational content. The discussions utilized a problem-solving approach and were individualized to the subject and, when available, caregiver needs [18]. This necessarily led to variations, as one would expect in a more pragmatic, clinically-oriented intervention. However, this allowed the intervention to be flexible to meet the varying needs of subjects and their caregivers.

Forty (19%) patients stopped the intervention and were lost to follow-up. Since we were not able to confirm whether these patients died, went to hospice, or formally withdrew, we had to treat them as censored in the analysis under the assumption that censoring was at random. It is impossible to formally assess this assumption. However, future studies may be able to consider these patients non-engagers in the same way as withdrawals for a more robust lost to follow-up sample size if death and hospice information is more accessible.

## 6. Conclusions

A telephonic nurse-led palliative care program demonstrated high levels of subject engagement, ACP completion, and hospice use prior to death. Whites and those with low symptom burden more frequently withdrew. More research is needed to identify why certain individuals are more or less likely to engage in nurse-led telephonic care, as well as whether prioritizing individuals with high-symptom burden and/or those closer to death is more effective in improving care for persons living with advanced cancer.

## Figures and Tables

**Figure 1 cancers-15-02310-f001:**
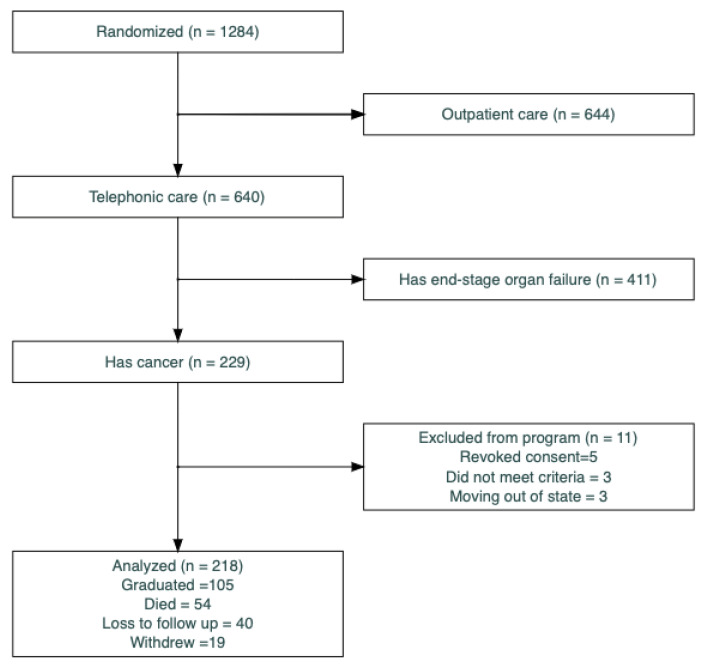
Consolidated Standards of Reporting Trials (CONSORT) diagram.

**Table 1 cancers-15-02310-t001:** Characteristics of study participants, showing the distribution among those that withdrew and did not withdraw from the program (*N* = 218).

Characteristic	Total (*N* = 218, %)	Not Withdrawn (*n* = 199, %)	Withdrawn (*n* = 19, %)
Age (mean (SD))	66 (10)	65 (9)	69 (12)
Male	107 (49)	94 (47)	13 (68)
Race			
White	159 (73)	142 (72)	17 (89)
Black	45 (21)	43 (22)	2 (11)
Other Race	13 (6)	13 (7)	0 (0)
Did not respond	1 (<1)	1 (<1)	0 (0)
Ethnicity			
Hispanic	12 (6)	11 (6)	1 (5)
Not Hispanic	204 (94)	186 (94)	18 (95)
Did not respond	2 (<1)	2 (1)	0 (0)
Functional Status (Missing = 1)			
Disabled	6 (3)	5 (3)	1 (5)
Requires considerable assistance	23 (11)	21 (11)	2 (11)
Requires occasional assistance	52 (24)	46 (23)	6 (32)
Cares for self, unable to do normal activity	51 (23)	49 (25)	2 (11)
Normal activity	84 (39)	76 (38)	8 (42)
Did not respond	1 (<1)	1 (<1)	0 (0)
Primary Language—English	215 (99)	196 (98)	19 (100)
Income (Missing = 1)			
Less than USD 25 K	58 (27)	54 (27)	4 (21)
USD 25 K–USD 49,999 K	36 (17)	33 (17)	3 (16)
USD 50 K–USD 99,999 K	35 (16)	31 (16)	4 (21)
USD 100 K or more	48 (22)	44 (22)	4 (21)
Did not respond	40 (18)	35 (18)	5 (26)
Education Level			
<High school degree	15 (7)	14 (7)	1 (5)
High school degree	53 (24)	48 (24)	5 (26)
Some college/AA degree	69 (32)	65 (33)	4 (21)
College degree or >	80 (37)	72 (36)	8 (42)
Did not respond	1 (<1)	0 (0)	1 (5)
Marital Status			
Married	113 (52)	102 (51)	11 (58)
Never married	29 (13)	28 (14)	1 (5)
Widow(er)	21 (10)	19 (10)	2 (11)
Separated	6 (3)	5 (3)	1 (5)
Divorced	31 (14)	28 (14)	3 (16)
Living with a partner	7 (3)	7 (4)	0 (0)
Other	5 (2)	4 (2)	1 (5)
Did not respond	1 (<1)	1 (<1)	0 (0)
Religion			
Do not practice/believe	83 (38)	77 (39)	6 (32)
Catholic	45 (21)	40 (20)	5 (26)
Protestant	28 (13)	24 (12)	4 (21)
Jewish	6 (3)	6 (3)	0 (0)
Other	54 (25)	50 (25)	4 (21)
Did not respond	2 (<1)	2 (1)	0 (0)
No primary family caregiver (Missing = 1)	81 (37)	72 (36)	9 (47)
Recruited before COVID-19 period (<Feb 2020)	93 (43)	83 (42)	10 (53)
FACT-G score at baseline (mean (SD)) (Missing = 1)	68 (18)	68 (18)	71 (24)
Low symptom burden (ESAS-r) at baseline (mean (SD)) (Missing = 1)	88 (41)	77 (39)	11 (58)
Very lonely (Three-Item Loneliness Scale) at baseline (mean (SD)) (Missing = 1)	33 (15)	31 (16)	2 (11)

**Table 2 cancers-15-02310-t002:** Cox Proportional Hazards Model to examine the relationship between baseline characteristics and time to withdrawal from the program (*N* = 211).

Predictor	Hazard Ratio
Age	1.03 (95% CI: 0.99, 1.07)
Sex	
Male	1.86 (0.85, 4.09)
Female	Reference category
Have caregiver	
No	1.78 (0.59, 5.35)
Yes	Reference category
Income	
<25k yearly income	1.16 (0.38, 3.54)
25k+ yearly income	Reference category
Race	
White	**3.74 (1.01, 13.8)**
Not white	Reference category
Ethnicity	
Hispanic	1.88 (0.24, 14.6)
Not Hispanic	Reference category
Functional Status	
Requires considerable assistance or more	1.85 (0.54, 6.37)
Requires occasional assistance or less	Reference category
Education	
≤High school	1.18 (0.44, 3.16)
>High school	Reference category
Recruited pre-COVID-19	
Yes	1.53 (0.48, 4.86)
No	Reference category
FACT-G at baseline	0.99 (0.97, 1.02)
Symptom burden (ESAS-r) at baseline	
Low symptom burden	**1.90 (1.02, 3.53)**
High symptom burden	Reference category

Seven subjects’ data excluded due to missingness.

**Table 3 cancers-15-02310-t003:** Advance care planning (ACP) milestones completed among study participants.

Milestone	Totals (*N* = 218, %)
One or more ACP items completed	182 (83)
HCP named	165 (76)
HCP informed	165 (76)
HCP form completed	142 (65)
AD conversation completed	106 (49)
AD wishes documented in Electronic Health Record (Epic)	109 (50)
AD wishes shared with HCP	63 (29)
RN communicated with oncologist	64 (29)
AD wishes shared with oncologist	57 (26)
Discussed hospice with patient	48 (22)
Patient enrolled in hospice	46 (21)
Patient followed up with outpatient palliative care services	31 (14)

ACP; Advance Care Planning: HCP; Healthcare Proxy: AD; Advance Directive: RN; Registered Nurse.

**Table 4 cancers-15-02310-t004:** Logistic Regression Model to examine the importance of baseline variables in predicting hospice use of those who died (*N* = 54).

Predictor	Odds Ratio (Adjusted)
Age	1.10 (0.99, 1.34)
Male sex (vs. female)	0.28 (0.01, 2.35)
White (vs. non-white)	4.13 (0.41, 82.1)
Education ≤ High school (vs. education > High school)	7.89 (0.93, 156)
FACT-G at baseline	**0.92 (0.80, 0.99)**
Low symptom burden (ESAS-r) at baseline (vs. high)	3.87 (0.35, 148)

FACT-G: Functional Assessment of Cancer Therapy-General; ESAS-r: Edmonton Symptom Assessment Scale-revised. Two subjects’ data excluded due to missingness.

## Data Availability

All data generated and/or analyzed during the current study or sub-study are available from the corresponding author upon reasonable request.

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
