# Peer review of "Engagement, Advance Care Planning, and Hospice Use in a Telephonic Nurse-Led Palliative Care Program for Persons Living with Advanced Cancer"

_cancers, 2023, doi:10.3390/cancers15082310_

Round 1
Reviewer 1 Report
This is a concise and clear manuscript concerning a nursing telephone intervention to promote ACP and hospice use in advanced cancer. This secondary analysis of RCT data focuses on issues of program withdrawal/engagement in the intervention arm only. Overall findings indicate good acceptability of the program (about half stayed in the program) and 80% of those who died were enrolled in hospice. Some comments to improve the paper:
1. Can the authors provide descriptive statistics about how many sessions were administered and their length over the 6 month intervention period?
2. In Tables 2 and 4, please indicate which effects were significant using an asterix to improve readability.
3. Figure 2 does not seem to be referenced in the main text. Please also clarify why age is depicted in Figure 2, which suggests this was a significant effect and entered as a binary variable, but is not discussed as such in the main text (c.f. line 175).
4. There may be an issue concerning the analysis which appears to collapse those who were lost to follow-up together with those who completed the program/died. The current analysis appears to compare the 199 non-withdrawals vs 19 withdrawals. However shouldn't those who were lost to follow-up (n=40) be considered non-engagers in the same way as withdrawals? If so, should there be another analysis comparing those who were lost to follow-up vs those who were not lost?
5. Line 247: please provide comparison percentages.
Author Response
Engagement, advance care planning, and hospice use in a telephonic nurse-led palliative care program for persons living with advanced cancer
Cancers Special Issue
Dear Editor and Reviewers:
We would like to thank the reviewers of our manuscript for the time and expertise they have shared in reviewing our manuscript. We have addressed each comment/request below, made significant changes to the analysis and interpretation of our results, and feel the manuscript is much improved after incorporating these suggestions. For clarity, we have copied and pasted the issues raised (in bold) and we responded directly below the comments (in italic). Any edits made directly on the manuscript are highlighted in track changes for ease of review, and we have uploaded a marked-up version and clean version of the manuscript. We are happy to continue to make any additional requested edits, and please let me know if there are any questions.
Sincerely,
Allison Cuthel, MPH
E: Allison.Cuthel@nyulangone.org
---------------------------------------------------------------------------------------------------------------------
Reviewers’ Comments
Comment
This is a concise and clear manuscript concerning a nursing telephone intervention to promote ACP and hospice use in advanced cancer. This secondary analysis of RCT data focuses on issues of program withdrawal/engagement in the intervention arm only. Overall findings indicate good acceptability of the program (about half stayed in the program) and 80% of those who died were enrolled in hospice. Some comments to improve the paper:
Response: Thank you for your comments – and your review of our manuscript.
- Comment: Can the authors provide descriptive statistics about how many sessions were administered and their length over the 6 month intervention period?
Response: This data was not collected throughout the course of the study period. We agree that this may be seen as a limitation, and is listed in Line 282.
- Comment: In Tables 2 and 4, please indicate which effects were significant using an asterisk to improve readability.
Response: We agree that readability will be enhanced if we highlight the factors where the 95% CI do not cross over OR=1. However, we are trying to avoid using language that refers to “statistical significance,” since that is not really appropriate in this type of analysis (for example, see Wasserstein, Ronald L., and Nicole A. Lazar. "The ASA statement on p-values: context, process, and purpose." The American Statistician 70, no. 2 (2016): 129-133). We feel that the asterisk is too closely associated with statistical significance. Instead, we have bolded the confidence intervals that do not cross one. If the reviewers are fully committed to using an asterisk, we will address further.
- Comment:
- Figure 2 does not seem to be referenced in the main text.
- Please also clarify why age is depicted in Figure 2, which suggests this was a significant effect and entered as a binary variable, but is not discussed as such in the main text (c.f. line 175).
Response: We agree with you that Figure 2 is more confusing than helpful, and we feel that removing the figure entirely eliminates some of that confusion without losing anything in the communication of the information.
- Comment: There may be an issue concerning the analysis which appears to collapse those who were lost to follow-up together with those who completed the program/died. The current analysis appears to compare the 199 non-withdrawals vs 19 withdrawals. However shouldn't those who were lost to follow-up (n=40) be considered non-engagers in the same way as withdrawals? If so, should there be another analysis comparing those who were lost to follow-up vs those who were not lost?
Response: This is a very important observation, and one we gave much consideration to as we conducted the analysis. We were unable to confirm whether lost to follow-up patients included patients who had died, gone to hospice, or effectively withdrew; therefore, we chose to categorize them separately and treated them as censored in the analysis since we did not know if they had withdrawn. We have added this text into the limitations section (lines 288-293), given that the results do change if we consider those lost to follow-up (n=40) as non-engagers in the same way as withdrawals.
“Forty (19%) patients stopped the intervention and were lost to follow-up. Since we were not able to confirm whether these patients died, went to hospice, or formally withdrew, we had to treat them as censored in the analysis under the assumption that censoring was at random. It is impossible to formally assess this assumption. However, future studies may be able to consider these patients non-engagers in the same way as withdrawals for a more robust lost to follow-up sample size, if death and hospice information is more accessible.”
- Comment: Line 247: please provide comparison percentages.
Response: We have added the comparison percentages (line 248).
Reviewer 2 Report
This study analyzed results of a palliative cancer care program conducted telephonically by nurses. Findings were a strong level of engagement and completion of the program by patients, especially among patients with more severe symptoms and black patients. However, patients lost to follow up or withdrawal accounted for over 25% of the sample, consisting mostly of white patients and patients with less severe symptoms. This raises issues of how missing data were handled in the data analysis. Please see the attached pdf file for further comments.

Author Response
Response to Reviewers – Reviewer 2
Engagement, advance care planning, and hospice use in a telephonic nurse-led palliative care program for persons living with advanced cancer
Cancers Special Issue
Dear Editor and Reviewers:
We would like to thank the reviewers of our manuscript for the time and expertise they have shared in reviewing our manuscript. We have addressed each comment/request below, made significant changes to the analysis and interpretation of our results, and feel the manuscript is much improved after incorporating these suggestions. For clarity, we have copied and pasted the issues raised (in bold) and we responded directly below the comments (in italic). Any edits made directly on the manuscript are highlighted in track changes for ease of review, and we have uploaded a marked-up version and clean version of the manuscript. We are happy to continue to make any additional requested edits, and please let me know if there are any questions.
Sincerely,
Allison Cuthel, MPH
E: Allison.Cuthel@nyulangone.org
---------------------------------------------------------------------------------------------------------------------
Reviewers’ Comments
Comment
- This study analyzed results of a palliative cancer care program conducted telephonically by nurses. Findings were a strong level of engagement and completion of the program by patients, especially among patients with more severe symptoms and black patients.
However, patients lost to follow up or withdrawal accounted for over 25% of the sample, consisting mostly of white patients and patients with less severe symptoms. This raises issues of how missing data were handled in the data analysis. Please see the attached pdf file for further comments.
Response: We agree that this is an important issue, and is a limitation of the study (which we now formally acknowledge in the text in lines 288-293, as described above). We were unable to confirm whether lost to follow-up patients included patients who had died, gone to hospice, or effectively withdrew; therefore, we chose to categorize them separately and treated them as censored in the analysis since we did not know if they had withdrawn.
- Comment: Line 150: Could you comment in more detail? Were data missing at random? You mentioned that patients with specific characteristics were more likely to drop out. Did this bias your findings? Did you consider multiple imputation?
Response: The statement on line 150 refers to a small number of patients (7) who were completely excluded from the withdrawal analysis because they were missing baseline covariate information that was required for the Cox proportional hazards model. This missing data does not refer to the 40 censored patients (lost to follow-up), who were included in the analysis until the point when they were lost to follow up. We have clarified this in the methods and results sections (lines 173 and 181-182). See response #4 to Reviewer 1 for comment on censoring at random.
- Comment: Line 231: In later cancer stages
Response: The sentence now reads “This led authors to include patients in later cancer stages and with higher symptom burden in subsequent trials”
Round 2
Reviewer 2 Report
The corrected information appears on line 150-151:
"Observations with missing data at base line were excluded from the analyses."
Also lines 162-163:
"There were 211 patients included in the withdrawal analysis, because 7 patients had missing covariate information at baseline."